# Natural Antimicrobials for *Listeria monocytogenes* in Ready-to-Eat Meats: Current Challenges and Future Prospects

**DOI:** 10.3390/microorganisms11051301

**Published:** 2023-05-16

**Authors:** Aaron R. Bodie, Corliss A. O’Bryan, Elena G. Olson, Steven C. Ricke

**Affiliations:** 1Meat Science and Animal Biologics Discovery Program, Department of Animal and Dairy Sciences, University of Wisconsin, Madison, WI 53706, USA; aaronb@werfoodsafety.com (A.R.B.); egolson2@wisc.edu (E.G.O.); 2Food Science Department, University of Aransas-Fayetteville, Fayetteville, AR 72701, USA

**Keywords:** RTE meats, *Listeria monocytogenes*, essential oils, organic acids, bacteriocins, bacteriophage

## Abstract

*Listeria monocytogenes*, an intra-cellular, Gram-positive, pathogenic bacterium, is one of the leading agents of foodborne illnesses. The morbidity of human listeriosis is low, but it has a high mortality rate of approximately 20% to 30%. *L. monocytogenes* is a psychotropic organism, making it a significant threat to ready-to-eat (RTE) meat product food safety. *Listeria* contamination is associated with the food processing environment or post-cooking cross-contamination events. The potential use of antimicrobials in packaging can reduce foodborne disease risk and spoilage. Novel antimicrobials can be advantageous for limiting *Listeria* and improving the shelf life of RTE meat. This review will discuss the *Listeria* occurrence in RTE meat products and potential natural antimicrobial additives for controlling *Listeria*.

## 1. Introduction

During 2019, FoodNet identified 25,866 cases of foodborne illness which had resulted in 6164 hospitalizations and 122 deaths [1]. Microorganisms such as *Salmonella*, *Listeria*, *Escherichia coli*, and *Campylobacter* are the primary etiological agents of gastrointestinal tract (GIT) foodborne illnesses in the United States [2]. The cross-contamination of foodborne pathogens can occur at any stage during food production. Accordingly, effective intervention strategies must continue to evolve to reduce human foodborne illness. Continued research and the development of intervention strategies are critical as retail markets demand change and subsequently impact food production technologies, including packaging. In addition, foodborne pathogens evolve and require different approaches for the concentrations and types of antimicrobials to sustain efficacy. Defining such multi-hurdle strategies associated with various pathogens remains a concern for food production in general, including for meat products.

Meat products can be exposed to foodborne pathogens at any production stage, from live animal operations to final retail handling and home preparation. Foodborne pathogens from the GIT of animals can cause cross-contamination in meat products [2]. For instance, in cattle, the hide and lymph nodes are significant contributors to pathogens such as *Salmonella*, which can increase the opportunity for cross-contamination [3]. In poultry, the feathers and crop can harbor pathogens such as *Listeria* that can result in the cross-contamination of carcasses [4]. However, contamination risk is not limited to cross-contamination during live production and processing. For example, pathogenic microorganisms can arise in food when directly preparing meals, handling the products, and through poor sanitation practices [5]. As a result, the contamination of meat products can end in significant morbidity and mortality worldwide [6]. The contamination possibility of meat products becomes especially important for Ready-to-Eat (RTE) meat products, where home preparation and handling can be a concern for foodborne pathogens such as *Listeria monocytogenes*. The challenge comes when the intervention choices are limited due to consumer requirements for only additives that are considered natural. The current review will discuss *Listeria* occurrence in RTE meat products and potential natural antimicrobial additives for controlling *Listeria* (Figure 1).

## 2. *Listeria monocytogenes*

*Bacterium monocytogenes* were first isolated by Murry et al. [7] in laboratory rabbits. A new name, *Listerella*, was proposed by Pirie [8] but was the same as a mycetozoan named in 1906; therefore, Pirie suggested *Listeria* as the genus, which was accepted [8]. *Listeria* is a Gram-positive, facultatively anaerobic, non-spore-forming bacteria with short rods approximately 0.5 µm by 2.0 µm with rounded ends [9]. The cells are curved in a “V” in single or short chains [10,11]. *Listeria* is commonly isolated in silage, soil, sewage, and water [12]. *Listeria* possesses multiple stress response mechanisms for overcoming varying temperatures (0 °C to 45 °C), salt concentrations (up to 10% NaCl), and pH levels (4.5 to 9.2) [13,14,15]. *Listeria* spp. are known to survive and grow in aqueous environments such as canals, ditches, rivers, and lakes, and can adapt to an ever-changing environment. The organism is psychrophilic but grows best at 37 °C with an optimum water activity of ≥ 0.97 and can grow at a pH as low as 4.4 [16]. These stress responses allow *Listeria* to survive and multiply in the adverse environmental conditions often present in food production facilities [17,18]. There are 30 species in the genus *Listeria*, of which only L. monocytogenes (LM) and *L. ivanovii* are currently considered human pathogens [19]. LM is highly pathogenic and zoonotic and is the species of most significant concern for food safety [20]. The consumption of meat contaminated with LM causes human foodborne listeriosis. Listeriosis is characterized by a high mortality rate of over 20% in infected individuals [21,22]. Although listeriosis accounts for less than 2% of foodborne illnesses in the United States, it has been responsible for 40% of the deaths caused by the disease. In addition, listeriosis results in the highest hospitalization rate (92% of infected individuals) compared to other foodborne illnesses [1]. Listeriosis causes approximately 1600 cases and approximately 260 deaths yearly [1]. This potentially fatal infection affects the GIT, leading to symptoms of nausea, diarrhea, vomiting, and headaches [23]. Persons most susceptible to listeriosis are the elderly, pregnant women, and the immunocompromised because their immune system is more vulnerable than that of healthy individuals [24]. The infective dose of LM is estimated to be anywhere from 10 to 100 million colony-forming units (CFU) in healthy hosts and only 0.1 to 10 million in immunocompromised individuals [25]. Furthermore, there is variation in the incubation periods, which depends on the mode of transmission and the dose received, but it typically ranges from one to four weeks and can be as high as several months [26].

Listeriosis outbreaks have been associated with many diverse food products, but the preponderance of cases arises from consuming RTE meat products (Table 1) [27]. Delicatessen meats have the highest predictive risk for causing listeriosis. These products can support the growth of LM under typical storage conditions and have a long shelf life compared to other RTE food products [27]. The contamination of RTE foods with LM causes approximately USD 2.8 billion in monetary losses in the US annually [28].

## 3. LM in RTE Meat

RTE foods, including red meat, poultry, seafood, and vegetables, have been documented as vehicles for several bacterial pathogens resulting in foodborne outbreaks [29,30]. The contamination of food by LM can occur at any point in the RTE meat processing chain. The transmission of LM in RTE meat occurs through cross-contamination during post-processing steps after the meat has been cooked [31]. Furthermore, LM can grow at temperatures as low as 2 °C, increasing the chance of consumers ingesting the pathogen [10]. In 1998, the largest LM in an RTE meat outbreak in the US occurred due to frankfurters, resulting in 101 hospitalizations, 15 deaths, and 6 miscarriages [32]. However, the largest known outbreak worldwide occurred in South Africa from 2017 to 2018, resulting in 1060 cases; of the 806 cases where a disease outcome is known, 216 died [33]. RTE meat products have the highest risk of deaths per serving and per annum among all other food products [34]. The frequency and increased concentration of LM in RTE meat products occur via recontamination of the product before its final packaging and handling at the retail stage or at home [35]. Equipment and surfaces that encounter RTE meats after the cooking step are commonly associated with cross-contamination [36]. LM can persist in food processing plants by forming biofilms on related surfaces [37]. Several studies have shown that LM can adhere to and form biofilms on food contact surfaces such as polyethylene, polyvinyl chloride, glass, and stainless steel [38,39,40].

A biofilm is a microbial population of cells attached to a surface or to each other, typically encased in an extracellular polymeric substance [41]. Forming a biofilm increases biological fitness and promotes long-term survival in stressful environments. Biofilm formation is often supported by the accumulation of food residues in specific niches, such as meat choppers or minced meat machines [42]. LM can develop into a biofilm on hydrophilic substances in less than 2 h at 37 °C [43]. Biofilms are challenging because they are difficult to remove [44]. Taormina and Beuchat (2002) demonstrated that LM reduction through treatment with sodium hypochlorite and heat was approximately 100 times lower in a biofilm than in planktonic cells [45]. Once a *Listeria* population forms a biofilm, it builds resistance against heat, antibiotics, sanitizing agents, and environmental conditions such as pH, water availability, and nutrient availability [46]. Biofilms are extremely difficult to remove because they are highly resistant to conventional cleaners and sanitizers [47]. This makes the prevention and control of biofilms of the utmost importance [48]. One method of biofilm prevention is modifying the materials used in food industries to prevent biofilms [48]. The first step in biofilm formation is adherence to a surface; if the surface can be modified by changing its morphology or its hydrophobicity, hydrophilicity, or electrical charge, it might be possible to limit adhesion to the surface [48]. For instance, Hsu et al. [49] used silica and alumina surfaces which induced changes in the morphology of *E. coli*, *Listeria innocua*, and *P. fluorescens* cells, making them less adherent to these materials. The concentration of metal ions in stainless steels can also affect adherence, with higher metal ion concentrations known to reduce biofilm formation for *S. Typhimurium*, as well as increasing sensitivity to chlorine disinfectants [50]. More effective disinfectants for removing biofilms are also being researched. Enzymes are considered environmentally friendly and can break down the matrix holding the biofilm together [48]. For instance, DNase enzymes have been shown to enable the removal of biofilms, as demonstrated for *C. jejuni* [51] or *L. monocytogenes* [52,53]. Proteases such as proteinase K [52], lipases [54], and carbohydrate-degrading enzymes such as β-glucanase and α-amylase [55] have also been studied. Multiple areas are being researched for the control and removal of biofilms in the food industry.

## 4. Antimicrobial Additives for LM Control in RTE Meats

Antimicrobials have been applied to foods for centuries. However, consumers now demand alternatives that they perceive as “natural.” RTE processed meat products without synthetic chemicals have become increasingly popular with consumers over the last few years [56]. Although thermal processing is the only reliable method to inactivate LM [57], it may reduce food quality. Natural and organically processed meats continue to experience popularity through the perceived safety and health benefits of foods free of chemical preservatives and pesticides [58]. An antimicrobial agent is a substance that either inhibits growth (bacteriostatic) or kills bacteria (bactericidal) that can persist on meat products throughout their shelf-life [59]. A comprehensive list of antimicrobials and treatments permitted in RTE meat and poultry products can be found in Safe and Suitable Ingredients Used in The Production of Meat and Poultry Products [60]. The following section will cover natural antimicrobials currently used in the RTE food industry to combat LM.

### 4.1. Bacteriocins

Bacteriocins are microbially synthesized peptides or small proteins with antimicrobial activity [61]. Bacteria that produce these bacteriocins have an immunity to their products by synthesizing enzymes that make them resistant to the produced bacteriocins [62]. Bacteriocins are considered safe for human consumption because they are digested rapidly by proteases in the human GIT [63]. There are four categories of bacteriocins: lantibiotics, unmodified peptides, large proteins, and circular peptides [64]. Generally, bacteriocins’ mode of action involves denaturing the cell membrane [64]. The bacterial cell surface, which is anionic, is a common target exploited by the cationic bacteriocins [65,66]. The hydrophobic surfaces of the bacteriocin cross the lipid bilayer of the cell surface and subsequently polymerize into complexes such as ion-selective pores, which cause the dissipation of the proton motive force, the depletion of intracellular ATP, leakage of the intracellular substrates, and eventual death [67,68]. Many bacteriocins use docking molecules such as lipid II or mannose permease to interact with the membranes [69]. Garvicin ML is a circular bacteriocin derived from *Lactococcus garvieae* DCC43 that utilizes the maltose ABC transporter and permease as receptors [70]. Class I bacteriocins are lantibiotics with (methyl)lanthionine residues (lanthionine and methyllanthionine) that form intramolecular thioether rings forming ‘wedge-like’ pores. In contrast, class II bacteriocins increase membrane permeability through a `barrel stave’ pore [71,72].

Lactic acid bacteria (LAB) are the primary sources of known bacteriocins (Table 2) [73]. LAB can be found in various matrices, such as decomposing plant material and fruits, dairy products, fermented meat and fish, cereals, beets, pickled vegetables, potatoes, sourdough, silages, fermented beverages, juices, and sewage. Additionally, LAB colonizes humans and animals [74,75,76]. The bacteriocins from food-grade LAB are non-toxic to humans, do not alter the nutritional properties of the food product, and have been demonstrated to be effective at concentrations as low as 5 mg/kg [77]. The most common bacteriocin in meat products is nisin, produced by Lactobacillus lactis. Nisin is generally regarded as safe (GRAS) and belongs to the lantibiotic group. Nisin has been shown to reduce populations of Gram-positive bacteria, such as *Listeria* [78]. Food products targeted for the use of bacteriocins or bacteriocin-like inhibitory substances include meat, fish, dairy, cereal, fruit, and vegetable beverage products and other foods usually stored under refrigeration. Research has also demonstrated nisin’s efficacy in reducing *Listeria* populations in RTE meat [79]. Frankfurters treated with nisin resulted in a 2.35-log reduction in LM [79]. Ruiz et al. [80] showed a 4-log reduction in *Listeria* using nisin on RTE vacuum-packed diced turkey. Bacteriocins’ additional advantage as ribosome-synthesized peptides is the potential for bioengineering strategies [81], which might enhance bioactivity and specificity against foodborne pathogens and food spoilage organisms. Bacteriocins may also be engineered for improved solubility, protease resistance, and pH tolerance, further augmenting their value and effectiveness as antimicrobials [82].

Additional bacteriocins offer multiple applications as antimicrobials for controlling *L. monocytogenes* in RTE meat products. Other than nisin, the bacteriocins most active against LM are garviecin LG34, bifidocin A, leucocin C-607, pediocin GS4, plantaricin LPL-1, or pediocin PA-1 or sakacins (Table 2) [20,83,84,85,86,87]. These Class IIa bacteriocins form hydrophilic pores on the cytoplasmic membrane of Gram-positive microorganisms, dissipating the transmembrane electrical potential, resulting in intracellular ATP depletion, and causing the leakage of ions, amino acids, proteins, and nucleic acids [87,88]. Ultimately, depending on the mechanism and whether their targets are different, more than one bacteriocin could be applied simultaneously to achieve a synergistic antimicrobial impact on *L. monocytogenes*.

Although bacteriocins can be directly added to RTE meat products, other means exist for effective delivery. For example, using edible films infused with bacteriocins can control LM in RTE meat products, where they gradually release bacteriocins onto the surface of the meat product during the entire shelf life [89]. Aymerich et al. [90] assessed the effects of polyvinyl alcohol films infused with Enterocin A on LM inhibition in vacuum-packed, dry-cured ham stored under refrigeration [90]. The authors observed an immediate 1-log reduction of LM compared to the control. Furthermore, LM reduction was fourfold higher during six months of storage at 8 °C [90]. Lungu and Johnson [91] used zein coatings containing nisin on frankfurters and determined that nisin reduced LM counts by 6.6 CFU/g during 28 days of refrigerated storage. Bacteria can counteract the antimicrobial activity of bacteriocins by shifting the membrane surface charge and membrane fluidity [65,72]. Unfortunately, several strains of LM have developed a certain degree of resistance against bacteriocins. In the case of nisin, LM produced changes in the membrane lipid composition [92] and phospholipid charges that confer resistance to nisin [93,94]. However, the mutation of LM target receptors such as the mannose phosphotransferase (Man-PTS) receptor for Class IIa bacteriocins can play a vital role in resistance [95]. Ultimately, a multiple-hurdle approach using combinations of bacteriocins or various bacteriocins with other antimicrobial treatments can reduce LM and prevent resistance [96]. To ensure the optimal efficacy, this depends on the respective mechanism(s) possessed by each antimicrobial being combined as part of a multiple intervention application.

**Table 2 microorganisms-11-01301-t002:** Classification of bacteriocins (adapted from Kumariya et al. [96]).

Class		Features	Example	Mechanism of Action	Producers
I	Ia	Lantibiotics (<5 kDa peptides containing lanthionine and β-methyl lanthionine)	Nisin	Membrane permeabilization by pore formation	*L. lactis*
	Ib	Carbacyclic lantibiotics containing labyrinthin and labionin	Labyrinthopeptin A1	Not known	*Actinomadura namibiensis*
	Ic	Sactibiotics (sulfur to alpha carbon-containing antibiotics)	Thuricin CD	Not known	*B. thuringiensis*
II	IIa	Small heat-stable peptides, synthesized in a form of precursor which is processed after two glycine residues, active against *Listeria*, have a consensus sequence of YGNGV-C in the N-terminal	Pediocin PA-1, sakacins A and P, leucocin A.	Membrane permeabilization by pore formation	*P. pentosaceus*, *P. acidilactici*, *Lactobacillus sakei*
	IIb	Two component systems: two different peptides required to form an active poration complex	Lactococcins G, plantaricin EF and plantaricin JK	Membrane permeabilization by pore formation	*L. lactis* subsp. *cremoris*, *Lb. plantarum*
	IIc	Circular bacteriocins	Gassericin A, enterocin AS-48, garvicin ML	Membrane permeabilization by pore formation	*L. gasseri*, *E. faecalis*, *L. garvieae*
	IId	Unmodified, linear, leaderless, non-pediocin-like bacteriocins	Bactofencin A, LsbB	Membrane permeabilization by pore formation	*L. salivarius, L. lactis* subsp. *Lactis*
III		Large molecules sensitive to heat	Helveticin M, helveticin J and enterolysin A	Membrane permeabilization by pore formation	*Lb. crispatus*, *L. helveticus*, *E. faecalis*

Nisin remains the only bacteriocin approved for food use by the US Food and Drug Administration (FDA) and the European Union [97]. Nisin has a long history of safe use in food products, and thus qualifies as generally recognized as safe (GRAS) by the FDA [88]. Nisin is heat stable, effective at low concentrations, does not change the flavor or appearance of food, does not induce resistance in target organisms, does not affect the normal intestinal flora, and can be easily detected and quantified [97]. Many other bacteriocins have been studied but are not yet fully characterized and therefore have not yet been approved for use in the food industry.

### 4.2. Essential Oils

Aromatic plants and their components have been investigated as potential bacterial growth inhibitors, and their antimicrobial properties have been linked to essential oils and other secondary plant metabolites [98,99]. Essential oils (EO) are natural volatile compounds extracted from different parts of an aromatic plant, including bark, leaves, flowers, and seeds [100]. These oils comprise terpenes, aldehydes, alcohols, esters, phenolics, ethers, and ketones [99]. EOs are characterized by two or three significant components at high concentrations (20 to 70%), while other components are present in trace amounts [101]. For example, carvacrol (30%) and thymol (27%) are the main components of origanum EO [101]. The effect of EOs may also be different on bacterial species. The Gram-positive bacterial cell wall structure allows hydrophobic molecules to easily penetrate the cells and act on the cell wall and within the cytoplasm. In contrast, Gram-negative bacteria are generally more resistant [102,103]. The antimicrobial effect of EOs is reportedly mainly due to the phenolic compounds they contain [99,104]. The mode of action for EOs may include damage to cytoplasmic membranes, protein denaturation, the coagulation of cytoplasm, and the depletion of the proton motive force [103]. The inhibitory effect of EO on LM has been documented in numerous studies. Pirbalouti et al. [105] evaluated the impact of three EO from *Thymus daenensis* Celak, *Thymbra spicata*, and *Satureja bachtiarica* on chicken frankfurters. The results of their study indicated that LM populations increased during seven and fourteen days of storage at 4 °C on control frankfurters but decreased on frankfurters when exposed to EO treatment [104]. Menon and Garg also examined the antibacterial effect of clove oil in meat at 30 °C and 7 °C [106]. At concentrations of 0.5% and 1%, clove oil restricted the growth of LM in minced mutton at both temperatures, with 1% being more efficacious [106]. Awaisheh et al. [107] showed that EOs slowed LM populations’ growth rates compared to the control during 14 days of storage at 4 °C. The authors tested the efficacy of EO (1% *v*/*w*) from fir and qysoom in a meat luncheon model against two levels of an LM cocktail (3- and 6-log CFU/g) coupled with storage at 4 °C for 14 days. At the end of the storage time, for samples with low contamination, fir, qysoom, and a mixture exhibited approximately 6.37, 6.04, and 5.53-log CFU/g of LM, respectively, compared to the control, which reached a level of 6.90-log CFU/g. In samples with high contamination levels, LM populations resulted in 8.43, 8.88, and 6.75-log CFU/g for fir, qysoom, and a mixture, respectively, compared to 9.90-log CFU/g for the control.

Additionally, when considering EO and related compounds as potential natural antimicrobial additives, sensory evaluation should be considered as a part of the screening process to ensure consumer acceptance. Dos Santos et al. [108] tested commercial EO from cinnamon, clove, oregano, ginger, thyme, and plant extracts of pomegranate, olive, acorn, strawberry tree, and dog rose for activity against LM in a dry-cured ham model. The strawberry tree and pomegranate extracts showed minimal anti-listerial activity, while 10% cinnamon oil inhibited LM growth in the ham model. However, using cinnamon in meat had a strong negative sensory impact [98]. Limitations of EOs in food are their strong organoleptic flavor, low water solubility, and low stability [109]. Avoiding these adverse sensory effects may require certain combinations to achieve synergistic antimicrobial efficacy and thus allow lower concentrations of individual compounds.

### 4.3. Spices and Herbs

Spices and herbs are plant-derived substances that add flavor to foods. Spices are derived from roots, rhizomes, stems, leaves, bark, flowers, fruits, and seeds, while herbs are typically considered non-woody plants [109]. Several studies have demonstrated that clove oil could inhibit LM and other foodborne pathogens [110,111,112]. Zhang et al. [112] found that clove, rosemary, cassia bark, and licorice extracts produced substantial anti-listerial activity. However, the mixture of rosemary and licorice extracts was the most effective. The authors observed a significant reduction in LM when rosemary and licorice extracts of 2.5, 5, and 10 mg/mL were sprayed on RTE ham slices inoculated with LM. The growth of LM over 28 days at 4 °C was reduced by 2.5-, 2.6-, and 3-logs CFU/cm^2^, respectively [112]. The antimicrobial portions of spices and herbs have been associated with flavonoids, phenolic acids, lignans, and polymeric tannins [113,114,115]. These compounds act by binding to and penetrating the bacterial cell membrane, generating pores that increase the permeability, especially in Gram-positive pathogens such as LM [116]. Phenolic acids, such as benzoic acid, cause hyper-acidification at the plasma membrane interphase, which alters cell membrane potential, changes its permeability, and affects the NaC/KC ATPase pump implicated in ATP synthesis [117].

### 4.4. Organic and Inorganic Acids

Organic acids are carboxylic acids that include fatty acids [118]. They are all found in nature, usually in foods such as fruits, vegetables, and fermented foods, but are often manufactured chemically for use in animal feeds and human foods [119,120,121]. The antimicrobial effect of organic acids is produced by the diffusion of the protonated acid through cell membranes, followed by intracellular dissociation, resulting in cytoplasm acidification and intracellular acid anion accumulation [119,120,121]. Factors that affect the antimicrobial activity of organic acids depend on pH, acid concentration, ionic strength, the acid’s molecular weight, the ratio of protonated to unprotonated forms, and the bacterial strains being attacked. Organic acids such as citric, acetic, lactic, and tartaric acids have exhibited bactericidal properties [122]. Lactic acid and diacetic acid have been previously described as anti-listeria compounds. Their salts (sodium lactate and sodium diacetate) are generally used because they possess optimal solubility at various pH levels [123].

Sodium lactate (SL), a GRAS chemical, is widely used as a preservative to prolong shelf-life and increase the safety of meat products. According to the International Association of Natural Product Producers (IANPP), SL is also considered a natural additive and labeled as an organic substance for use as an antimicrobial and processing aid [124]. Sodium lactate inhibits the growth of bacteria by reducing the water activity of food products, delaying the development of bacteria, and acidifying the intracellular pH [119]. According to Hwang and Juneja, [125] SL reduced the bacterial population of *E. coli*, *Salmonella*, and *Listeria* at refrigerated and “abused” temperatures. Combining the two acids yielded maximum inhibitory effects on LM [126]. Consequently, they are generally applied as a combination in commercial RTE meat and poultry products, such as in packaged luncheon meat slices, wieners, smoked-cooked ham, light bologna, and salami [127]. The maximum permissible levels of sodium diacetate and lactate in RTE meats, as stipulated by FSIS, are 0.25% and 4.8%, respectively. The most common levels are 0.125 to 0.25% sodium diacetate with 1.5–3% sodium/potassium lactate [128]. SL and sodium diacetate solutions reduced LM populations by 0.6- to 1.0-log CFU/cm^2^ over 28 to 40 days without visual changes in the frankfurters [123]. Glass et al. [129] determined that a combination of 1.6% SL and 0.1% sodium diacetate inhibited the growth of LM inoculated on cured ham slices subsequently stored at 4 °C for up to 12 weeks. Porto et al. [130] observed 5.1- and 5.4-log reductions of LM in frankfurters formulated with 2% and 3% potassium lactate after storage at 10 °C for 60 days compared with the control. SL (1.8%) used alone in frankfurter formulations inhibited the growth of LM for 50 days, and more so when combined with 0.25% sodium diacetate at a pH below 7.0. LM was inhibited throughout 120 days of refrigerated storage [131].

Bodie et al. [132] evaluated the bactericidal effect of GRAS compound, sodium bisulfate (SBS), and nisin on LM in frankfurters. Frankfurters were inoculated with LM and treated with water, SBS (0.75 and 1.5%), nisin (0.5, 1, and 2%), and combinations (0.75% SBS + 0.5% nisin, 0.75% SBS + 1% nisin, 1.5% SBS + 1% nisin, and 1.5% SBS + 2% nisin). The combination of SBS and nisin exhibited the most significant LM reduction. In addition, the authors concluded that using these antimicrobials did not alter the appearance of wieners. In a subsequent study, Bodie et al. [133] evaluated the effects of the GRAS compound SBS, SL, and their combination as antimicrobial dips to reduce LM on inoculated organic frankfurters. Frankfurters were treated with tap water or various combinations of SBS and SL by dipping for 10 s in the solution. After treatment, frankfurters were vacuum packaged and stored at 4 °C. An interaction of treatment and time was observed among the microbiological plate data with all experimental treatments reducing the growth potential of LM across time. The efficacy of treatments was inconsistent across time; however, on day 21, SBS (0.39%)-treated franks had the lowest growth potential compared to the control, suggesting that the use of SBS over SL is a more effective antimicrobial for the management of LM. To determine the impact of these treatments on the microbiota associated with the organic frankfurters, they also used 16S rRNA gene sequencing to characterize the microbiome [133]. They concluded that the treatments appeared to stabilize the non-*Listeria* population, suggesting that antimicrobials can influence shelf life. In the future, balancing the inhibition of pathogens on RTE meat products versus any potential adverse effects on shelf life will be an essential consideration for evaluating natural antimicrobials and their combinations. Microbiome analyses offer a more comprehensive approach to assessing the impact on microbial populations when specific antimicrobials are introduced. Such results can provide potential data inputs when modeling shelf-life predictions for these compounds.

Sodium levulinate, used as a flavoring agent in many foods, is another organic acid with GRAS status and an effective anti-listerial agent. It has been shown to inhibit the growth of LM more effectively than SL or the combination of SL (1.875%) and sodium diacetate (0.125%) in cooked turkey rolls and bologna [128]. Furthermore, adding 2% or more sodium levulinate to the turkey rolls and 1% or more sodium levulinate to bologna completely prevented the growth of LM during 12 weeks of refrigerated storage without altering the flavor profile of either product compared to the control [128]. Palumbo et al. [134] and Houtsma et al. [135] showed that using sodium chloride and nitrite increased the anti-listerial effect of organic acids or their salts. Thus, applying organic acids in cured meats such as ham, frankfurters, and bologna may significantly affect LM reduction. Ivy et al. [136] concluded that refrigeration temperature is essential for achieving anti-listerial activity when using organic acids. These findings indicated that multi-hurdle approaches should consider time and temperature to achieve synergistic antimicrobial effects on LM populations in RTE meats.

Using organic acids can provide a safe alternative for antimicrobials with a reduced impact on human health. Most essential applications of organic acids are linked with preserving RTE products to prevent contamination during post-processing [137]. RTE meat products are typically formulated, dipped, or sprayed with an organic acid. Surface application can be more effective than adding to the formulation since bacterial contamination often occurs at the product’s surface [137]. Additionally, since a small amount of the antimicrobial can produce a significant antimicrobial effect, no additional changes in food product formulations are required.

### 4.5. Bacteriophage

Bacteriophages (phages) are viruses that infect bacterial cells. Each phage targets a specific genus, serotype, or strain of bacteria [138]. Bacteriophages are found universally in nature and have been identified in soil, water [138], and food products, including meat, dairy, and vegetables [139,140,141,142,143]. The US Food and Drug Administration has approved two bacteriophage preparations, using phages P100 and LMP-102, as food ingredients to control LM [144]. Guenther et al. [145] determined that P100 reduced LM population levels by up to 5-logs in hot dogs, sliced turkey meat, and smoked salmon. Chibeu et al. [146] showed that LISTEX™P100 effectively reduced LM during storage at 4 °C with initial reductions in LM of 2.1-log_10_ CFU/cm^2^ and 1.7-log_10_ CFU/cm^2^ for cooked turkey and roast beef, respectively. At the end of the 28-day storage period, LM counts were reduced by approximately 2-logs CFU/cm^2^ compared to the control [146]. *Listeria* bacteriophage A511 was examined for its activity against LM internally or on the surface of a vacuum-packed cooked meat model [147]. Applying the phage directly to the surface of meat reduced the LM counts below detection limits, and they remained significantly lower than the controls for up to 20 days of refrigerated storage. However, when LM was inoculated inside the meat, the surface application of A511 did not successfully reduce the pathogenic counts. Bacteriophage application inside the meat product did not reduce LM counts sufficiently, whether LM was in or on the surface of the meat [137]. These findings support the idea of multi-hurdle approaches, which combine different antimicrobials with phages to produce the most effective LM reduction across various meat matrices. Utilizing immobilized phage cocktails on cellulose membranes against *Listeria* or *Escherichia coli*, Anany et al. [148] demonstrated the successful control of LM and *E. coli* O157:H7 in RTE meat under different storage temperatures and packaging conditions. Immobilized phages offer an opportunity to provide a more stabilized delivery system for limiting *Listeria* in packaged products and given the highly specific nature of these phages they are unlikely to alter the non-*Listeria* microbial populations.

Phages alone are likely not a solution for LM contamination in food. However, phages’ effectiveness significantly improves when used with other anti-listerial factors. Depending on the approach for application, phages may still be beneficial for food producers and consumers. A critical element will be to conduct further searches for additional anti-*Listeria* bacteriophages that possess properties eliciting optimal antimicrobial activities during RTE meat production and retail conditions, such as refrigeration. Likewise, identifying bacteriophages that retain maximum lytic activity in the presence of other antimicrobials, such as essential oils or organic acids, would be interesting for multiple hurdle applications. Once new phages have been identified, the further development of any new bacteriophage-based LM inhibition in RTE meats will necessitate determining the performance, safety, and stability of phages in these meat products, and studies to ensure product consumption safety for retail meats.

There are still several challenges to using bacteriophages in food applications. For instance, bacteriophages are specific to their target organism, which requires identifying and selecting phages for each organism targeted [149,150]. It is also possible for the target bacteria to develop resistance to the phage over time [149,150]. Each bacteriophage must also be evaluated for its effects on human health and animal welfare, as well as for its effects on the environment [149,150]. Finally, there is the issue of consumer acceptance and whether the consumer will accept the phages as natural or believe them to be genetically modified organisms [149,150].

## 5. Conclusions

Consumer demand for convenient, taste appealing, and wholesome RTE meat and poultry is currently high. However, the susceptibilities of RTE products to the contamination and growth of LM pose consistent health risks to consumers. The food industry is constantly striving for better technologies to combat LM in RTE meat products, which are particularly susceptible due to the ubiquitous nature of LM, the long shelf-life of RTE meats, and the risk that consumers do not adequately handle and store these products. This review has presented the latest research on and briefly discussed the current state of natural anti-listerial compounds. These biological antimicrobial agents provide an affordable and valuable intervention strategy. However, efforts must continue in this critical and active research field to develop effective new technologies and pertinent regulatory guidelines to ensure the safest possible supply of RTE products.

With advanced diagnostic tools such as 16S rRNA gene sequencing, natural antimicrobials can be evaluated not only from the context of inhibiting *Listeria*, but simultaneously to determine if any unintended consequences for the non-*Listeria* microbial population may occur in RTE meat products during storage. These findings could harm economic outcomes, such as a shortened shelf-life for RTE meat products. Using whole genome sequencing on LM populations can help delineate individual strain differences and their responses to antimicrobial methods. With this knowledge, multi-hurdle approaches can be designed to be more effective across a broader spectrum of LM.

## Figures and Tables

**Figure 1 microorganisms-11-01301-f001:**
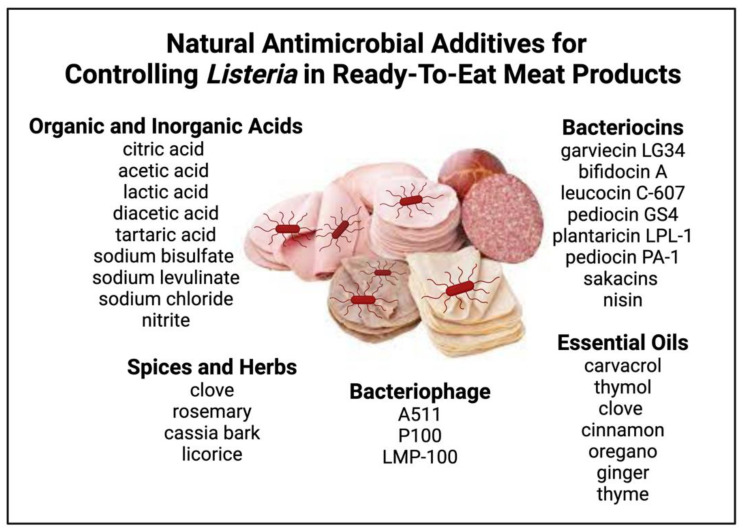
Natural antimicrobial additives for controlling *Listeria* in ready-to-eat meat products. Figure created with Biorender.com (accessed on 15 May 2023).

**Table 1 microorganisms-11-01301-t001:** Reported listeriosis outbreaks from RTE meats by year, Foodborne Disease Outbreak Surveillance System, United States, 1998–2022. https://www.cdc.gov/foodsafety/outbreaks/lists/outbreaks-list.html, accessed on 12 May 2023.

Study Period, Year †	Multistate	Total No. Cases ‡	No. Hospitalizations	No. Deaths	Food Vehicle
1998	Yes	108	101	14	Frankfurters
	No	4	NA *	NA	Frankfurters
	No	4	NA	NA	Frankfurters
	No	5	5	1	Deli meat
	Yes	11	NA	NA	Pâté
	No	2	2	1	Deli meat
	Yes	30	29	4	Deli meat
2001	No	28	0	0	Deli meat
2002	Yes	54	NA	8	Deli meat
	No	3	3	0	Grilled chicken
	Yes	13	13	1	Deli meat
2010	No	14	7	2	Hog head cheese
2016	Yes	10	10	1	Deli sliced meat
2018	Yes	4	4	1	Deli ham
2020	Yes	12	12	1	Deli meat
2021	Yes	3	3	1	Cooked chicken
2022	Yes	16	13	1	Deli meat

* Data not available. † No listeriosis outbreaks from RTE meat were reported in 2003–2009, 2011–2015, or 2019. ‡ Includes laboratory-confirmed and epidemiologically linked cases.

## Data Availability

Not applicable.

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
