# Peer review of "Natural Antimicrobials for Listeria monocytogenes in Ready-to-Eat Meats: Current Challenges and Future Prospects"

_microorganisms, 2023, doi:10.3390/microorganisms11051301_

Round 1

Reviewer 1 Report

Dear authors, the review is a relevant subject with merit to be published, however, I recommend a review for the following observations:

Topic 1

Line 41: The reference does not correspond to object.

Topic 2

Line 62-63: excessive references about the characteristic of microorganisms.

Line 72: “There are 21 species in the genus Listeria, of which only L. monocytogenes (LM) 72 and L. ivanovii are currently considered human pathogens [19]. Update in http://www.bacterio.net/listeria.html and quote the date

Line 80: Listeriosis causes approximately 1600 cases and approximately 260 deaths 80 yearly [1]. Update the reference.

Table 1: reanalyse the table data:

2016 There were no cases.;

2019 https://www.cdc.gov/listeria/outbreaks/deliproducts-04-19/index.html

Fix - No listeriosis outbreaks from RTE meat were reported in 2003-2009, 2011-2015, 2019.

2018; Deli ham No. hospitalizations; 4

Cite the sources in the table: Between 2016-2022 - https://www.cdc.gov/foodsafety/outbreaks/lists/outbreaks-list.html

Between 1998-2010 – cite the source

Line 100: I suggest continuing on the topic in topic 2 and in topic 3 talking briefly about biofilm and control by sanitizers to then enter topic 4. 

I missed addressing about: nanotechnology agents against L. monocytogenes; legal aspects and microbiological criteria for Listeria spp. in the food processing chain and RTE meat and legislation and regulation for the additives addressed in the review.

Author Response

Reviewer 1:

Dear authors, the review is a relevant subject with merit to be published, however, I recommend a review for the following observations:

Topic 1

Line 41: The reference does not correspond to object. We disagree. Line 41: For instance, in cattle, the hide and lymph nodes are significant contributors to pathogens such as Salmonella, which can increase the opportunity for cross-contamination [3]. Reference: 3. Arthur, T.M., D. M. Brichta-Harhay, J. M. Bosilevac, M. N. Guerini, N. Kalchayanand, J. E. Wells, S. D. Shackelford, T. L. Wheeler, and M. Koohmaraie. Prevalence and characterization of Salmonella in bovine lymph nodes potentially destined for use in ground beef. 2008. J of Food Prot. 71, 1685–1688.

Topic 2

Line 62-63: excessive references about the characteristic of microorganisms.

Authors believe this information is pertinent

Line 72: “There are 21 species in the genus Listeria, of which only L. monocytogenes (LM) 72 and L. ivanovii are currently considered human pathogens [19]. Update in http://www.bacterio.net/listeria.html and quote the date Corrected

Line 80: Listeriosis causes approximately 1600 cases and approximately 260 deaths 80 yearly [1]. Update the reference.

Updated: During 2019, FoodNet identified 25,866 cases of foodborne illness, resulting in 6,164 hospitalizations, and 122 deaths [1]. 1.    Tack, D.M., Ray L., Griffin P.M., Cieslak P.R., Dunn J., Rissman T., Jervis R., Lathrop S., Muse A., Duwell M., Smith K., To-bin-D'Angelo M., Vugia D.J., Zablotsky Kufel J., Wolpert B.J., Tauxe R., Payne D.C. Preliminary incidence and trends of in-fections with pathogens transmitted commonly through food - Foodborne Diseases Active Surveillance Network, 10 U.S. Sites, 2016-2019. MMWR Morb Mortal Wkly Rep. 2020 May 1;69(17):509-514. doi: 10.15585/mmwr.mm6917a1. PMID: 32352955; PMCID: PMC7206985.

Table 1: reanalyse the table data:

2016 There were no cases.; This outbreak began in 2016 and was closed in 2019.

2019 https://www.cdc.gov/listeria/outbreaks/deliproducts-04-19/index.html

Fix - No listeriosis outbreaks from RTE meat were reported in 2003-2009, 2011-2015, 2019. It is correct as stated

2018; Deli ham No. hospitalizations; 4 corrected

Cite the sources in the table: Between 2016-2022 - https://www.cdc.gov/foodsafety/outbreaks/lists/outbreaks-list.html Added

Between 1998-2010 – cite the source

Unsure what this is in reference to

Line 100: I suggest continuing on the topic in topic 2 and in topic 3 talking briefly about biofilm and control by sanitizers to then enter topic 4. Biofilm information added

Biofilms are extremely difficult to remove because they are highly resistant to conventional cleaners and sanitizers [47]. This makes prevention and control of biofilms of the utmost importance [48]. One method of biofilm prevention is modifying the materials used in food industries to prevent biofilms [48]. The first step in biofilm formation is adherence to a surface; if the surface can be modified by changing its morphology or its hydrophobicity, hydrophilicity, or electrical charge it might be possible to limit adhesion to the surface [48]. For instance, Hsu et al. [49] used silica and alumina surfaces. Which induced changes in the morphology of E. coli, Listeria innocua, and P. fluorescens cells change their morphology and perhaps be less adherent to these materials. The concentration of metal ions in stainless steels can also affect adherence with higher metal ion concentrations known to reduce biofilm formation by S. Typhimurium, as well as an increased sensitivity to chlorine disinfectants [50].  More effective disinfectants for removing biofilms are also being researched. Enzymes are considered environmentally friendly and can break down the matrix holding the biofilm together [48]. For instance, DNase enzymes have been shown to enable the removal of biofilms, as demonstrated for C. jejuni [51] or L. monocytogenes [52, 53]. Proteases such as proteinase K [52], lipases [54], and carbohydrate-degrading en-zymes such as β-glucanase and α-amylase [55] (Araújo et al. 2017) have also been studied. Multiple areas are being studied for the control and removal of biofilms in the food industry.

I missed addressing about: nanotechnology agents against L. monocytogenes; legal aspects and microbiological criteria for Listeria spp. in the food processing chain and RTE meat and legislation and regulation for the additives addressed in the review.

Authors believe that these topics are outside the scope of the article

Reviewer 2 Report

The manuscript is a well written review of the use of various natural antimicrobials in the control of Listeria monocytogenes in RTE meat products. However, the title suggests a review of current challenges and future prospects but some aspects are not fully addressed for bacteriocins and bacteriophage. The reasons why nisin remains the only approved bacteriocin is not discussed nor the challenges for the use of bacteriophage.

Author Response

The manuscript is a well written review of the use of various natural antimicrobials in the control of Listeria monocytogenes in RTE meat products. However, the title suggests a review of current challenges and future prospects but some aspects are not fully addressed for bacteriocins and bacteriophage. The reasons why nisin remains the only approved bacteriocin Added: Nisin remains the only bacteriocin approved for food use by the US Food and Drug Administration (FDA) and the European Union [88]. Nisin has a long history of safe use in food products and thus qualifies as generally recognized as safe (GRAS) by the FDA [88]. Nisin is heat stable, effective at low concentrations, does not change the flavor or appearance of food, does not induce resistance in target organisms, does not affect the normal intestinal flora, and can be easily detected and quantified [88]. Many other bacteriocins have been studied but are not yet fully characterized and therefore have not yet been approved for use in the food industry.

 is not discussed nor the challenges for the use of bacteriophage.

We have added: There are still several challenges to using bacteriophages in food applications. For instance, bacteriophages are specific to their target organism, which requires identifying and selecting of phage for each organism targeted [149,150]. It is also possible for the target bacteria to develop resistance to the phage over time [149, 150]. Each bacteriophage must also be evaluated for its effects on human health and animal welfare as well as for its effects on the environment [149, 150]. Finally, there is the issue of consumer acceptance and whether the consumer will accept the phages as natural or believe them to be genetically modified organisms [149, 150].